# A MISSING TESTBED FOR LLM PRE-TRAINING MEMBERSHIP INFERENCE ATTACKS

## ABSTRACT

We introduce a simple and rigorous testbed for membership inference attacks (MIA) against pre-training sequences for large language models (LLMs). Our testbed addresses the following gaps in existing evaluations, which lack: (1) *uniform* sampling of member/non-member documents of varying lengths from pre-training shards; (2) large-scale *deduplication* at varying strengths, both within and across the sampled members/non-members; and (3) rigorous *statistical tests* to detect member/non-member distribution shifts that cause faulty evaluations and are otherwise imperceptible to the heuristic techniques used in prior work. We provide both global- and domain-level datasets (e.g., Reddit, Stack Exchange, Wikipedia), derived from fully-open pre-trained LLM/dataset pairs including Pythia/Pile, Olmo/Dolma, and our custom pre-trained GPT-2-Large on FineWeb-Edu. We additionally open source a modular and extensible codebase that facilitates the creation of custom, statistically validated, and deduplicated evaluation data using future open models and datasets. In sum, our work is a concrete step towards addressing the evaluation issues discussed by prior work.

## 1 INTRODUCTION

We are interested in the rigorous evaluations of membership inference attacks (MIAs) algorithms against the pre-training data of large language models (LLMs). MIAs are binary classification algorithms to determine whether a given example $x$ was used for training for some model $M$ (i.e., whether $x$ is a *member* of $M$'s training set). With the surge of LLMs, there has been growing interest in developing MIA methods against these models for the declining levels of data transparency and the need for auditing data misuse (e.g., Shi et al. (2023); Zhang et al. (2024a); Li et al. (2023); Ko et al. (2023); Chang et al. (2024); Mattern et al. (2023)).

Despite this growing interest, progress has been limited. The key challenge is that very little is known about the training data of recent LLMs such as Llama (Dubey et al., 2024) and DeepSeek (Liu et al., 2024); this means that an MIA developer must first acquire a set of *ground truth* member (and non-member) sequences that are trained (and not trained) by this model to test their MIA algorithm.

This data acquisition process is where errors—specifically *distribution shifts* between members and non-members used for evaluation—occur in recent literature. For example, past work resorted to using *temporal thresholding*, where Wikipedia articles (Shi et al., 2023), arXiv papers (Duan et al., 2024), and books (Meeus et al., 2024a) before/after a training cut-off date are used as members/non-members. Subsequently, Duan et al. (2024); Das et al. (2024); Meeus et al. (2024d); Maini et al. (2024) reported that such temporal thresholding introduced distribution shifts so large, that even *model-blind* techniques like bag-of-words classifiers can outperform state-of-the-art MIA methods; in fact, the EMNLP 2024 best paper award paper on membership inference (Zhang et al., 2024b) was challenged due to this very concern (Maini & Suri, 2024). Another line of work considers using model generations as synthetic non-members (e.g., Kazmi et al. (2024); Guépin et al. (2023)), and Naseh & Mireshghallah (2025) found that the synthetic text distribution diverges from the training distribution sufficiently such that the MIAs essentially become machine-generated text classifiers. To further illustrate the repercussions of member/non-member distribution shifts, we investigated recent work claiming high MIA performance, only to find that the proposed method overfitted to the extra EOS tokens in non-member (but not member) sequences.[1] These instances suggest an urgent need for rigorous datasets for MIA evaluations.

---

[1] After we communicated our findings, the authors diligently retracted the paper.

Figure 1: Illustration of our pipeline to create a dataset of verified members and non-members. We repeat for different model/dataset pairs, deduplication strengths, and global-/domain-level datasets.

Our work aims to bridge this gap. We provide a simple testbed using *fully-open* model/dataset pairs (e.g., Olmo (Groeneveld et al., 2024) and Dolma (Soldaini et al., 2024)) such that ground truth members and non-members are known and that true uniform samples can be drawn without distribution shifts. We then apply large-scale deduplication both *within* and *across* the sampled members and non-members to minimize overlap which confounds evaluation. In particular, the non-members are also deduplicated against the *entire* pre-training set. Importantly, deduplication inevitably introduces member/non-member distribution shifts for natural text (there will be non-zero shift as long as some text appears in one set but not the other). To rigorously monitor distribution shifts stemming from both sampling and deduplication, we implement two-sample Kolmogorov–Smirnov (KS) tests over a range of textual features (e.g., average word lengths, vocab richness) across a spectrum of deduplication strengths (e.g., different percentages of $n$-gram overlap between two sequences).

**In sum, our work introduces a concrete testbed that captures the key desiderata discussed, yet not implemented, in prior work (see Appendix B for related work)**:

1. True *uniform* sampling of member/non-member sequences across pre-training shards;
2. Large-scale *deduplication* at varying strengths; and
3. Rigorous *statistical tests* to quantify member/non-member distribution shifts.

**Summary of artifacts.** We consider three model/dataset pairs: (1) **Pythia/Pile** (Biderman et al., 2023; Gao et al., 2020), (2) **Olmo/Dolma** (Groeneveld et al., 2024; Soldaini et al., 2024), and (3) our own custom pre-trained **GPT-2/FineWeb** (Radford et al., 2019; Penedo et al., 2024). For each model-dataset pair, we release both global- and domain-level (e.g., Reddit, Stack Exchange, Wikipedia) member and non-member sets that are pre-sampled, deduplicated, and statistically tested. For Pythia/Pile and GPT-2/FineWeb, we also release a list of datasets corresponding to a range of deduplication strengths (§2.3), as the strength trades off sample overlap against distribution shifts. Finally, we open source a modular, extensible, and easy-to-use toolkit to reproduce these artifacts and to create custom evaluation datasets for future open model/dataset pairs.

## 2 METHODS

### 2.1 DATASET CONSTRUCTION

Modern LLMs are trained on vast datasets such that it is unclear what data are guaranteed to be excluded unless explicitly stated by the model developer. Thus, a key requirment for a reliable model/dataset pair is the existence of an official *validation* set with a similar distribution as the training set. Despite the growing number of open models, few currently meet this criterion: open-weight models like Llama (Dubey et al., 2024) and DeepSeek (Liu et al., 2024) have proprietary training data, while many fully open-source models like LLM360 (Liu et al., 2023) and DCLM (Li et al., 2024) do not have official validation sets. Our testbed uses the following model/dataset pairs:

• **Pythia/Pile**: The official Pile dataset (Gao et al., 2020) has established train-validation split, and the Pythia suite of models (Biderman et al., 2023) has fully documented the training procedures such that the ground-truth members and non-members are available.[2]

• **Olmo/Dolma**: Similar to Pythia, the Olmo family of models (Groeneveld et al., 2024) is fully open-source, and Dolma (Soldaini et al., 2024) was released as the official training set. Unlike Pile, Dolma does not have an "official" train-validation split; however, Paloma Magnusson et al.

---

[2]Note that the original Pile is no longer accessible due to copyright concerns. We use the "uncopyrighted" version available at https://huggingface.co/datasets/monology/pile-uncopyrighted.

(2023) was used as the official evaluation data and the Olmo developers applied decontamination to minimize the overlap against Dolma. We thus sample non-member data from Paloma.

- **Controlled GPT-2/FineWeb**: To establish a fully controlled environment, we trained a GPT-2-Large model *from scratch* on the FineWeb-Edu dataset (Penedo et al., 2024) to Chinchilla-optimal ($\approx 20$ tokens per parameter, Hoffmann et al. (2022)). We release checkpoints after one, two, and three pre-training epochs, providing complete transparency about the training data.

Our testbed incorporates two key design choices to ensure realistic assessment of MIA. First, we use complete documents rather than fixed-length segments as our basic evaluation units, better reflecting real-world scenarios where entire documents are used in pretraining. Second, we evaluate on both global and domain-specific subsets (e.g., Reddit, Stack Exchange), enabling fine-grained analysis across different data distributions. See Appendix C for detailed discussion of these design choices.

## 2.2 TRUE AND EFFICIENT UNIFORM SAMPLING

We implement true random sampling for terabyte-scale sharded datasets, addressing the computational challenges noted in Duan et al. (2024). Our scalable framework performs a one-time preprocessing to create shard indices (e.g., recording newline offsets for fast document-level indexing) and collect metadata (e.g., document domains), optimized through multi-threaded execution. The shard indices enable both $O(1)$ unconditional random sampling from the *entire* pre-training dataset and conditional random sampling based on document domains, eliminating the need for costly linear scans of massive datasets. More details can be found in Appendix C.

## 2.3 LARGE-SCALE DEDUPLICATION

**Definition and strength of deduplication.** Pre-training datasets are large and thus documents are possibly duplicated both within and across the train and validation splits. This makes deduplication a critial step for proper evaluation. Following Anil et al. (2023), we define a document as a duplicate if $\geq d\%$ of its 8-grams appear at least once in the training corpus, where different values of $d\%$ captures varying deduplication strengths (lower means stronger deduplication). We use $d = 70$ by default, though we may provide sampled datasets across all strengths. For a detailed analysis of how deduplication strength affects distribution shifts and membership inference, see Appendix G.

**Inter-deduplication.** We first deduplicate *across* the sampled members and non-members, i.e., ensuring that a document in one set has no duplicate in the other set. Note that this involves deduplicating non-members against the *entire* pre-training set (not just the sampled members), since we need to ensure none of the sampled non-members were seen by the model. For efficiency, we follow Groeneveld et al. (2023) for non-member deduplication using a large-scale *bloom filter* of training $n$-grams, which may over-deduplicate but does not affect correctness.

**Intra-deduplication.** As a second stage of decuplication, we also deduplicate documents *within* each sampled set of members/non-members to prevent biased evaluation scores. This similarly leverages an efficient bloom filter implementation for rapid $n$-gram overlap checks.

## 2.4 DETECTING MEMBER/NON-MEMBER DISTRIBUTION SHIFTS

We implement two complementary approaches to detect distribution shifts between members and non-members. First, following Das et al. (2024); Meeus et al. (2024b), we use "blind" baselines that distinguish members vs. non-members without accessing the model; blind baselines, by construction, should perform randomly on ideal evaluation data. Second, we conduct rigorous statistical analysis using Kolmogorov-Smirnov (KS) tests on a comprehensive set of textual features (e.g., word count, vocabulary richness) to quantify distribution shifts. The KS test statistic measures the magnitude of shifts, while its $p$-value indicates confidence given the sample size, providing more sensitive detection than blind baselines. See Appendix H for detailed methodology.

## 3 DATASET ANALYSIS

We applied our pipeline described in §2 to the three model/dataset pairs (Pythia/Pile, Olmo/Dolma, and GPT-2/FineWeb). This section analyzes the quality of these evaluation datasets. We summarize the source datasets in Table 3 in the Appendix.

| Duplicate Strength | 100% | | 90% | | 70% | | 50% | | 30% | | 10% | |
|---|---|---|---|---|---|---|---|---|---|---|---|---|
| Features | KS | p-value | KS | p-value | KS | p-value | KS | p-value | KS | p-value | KS | p-value |
| word_count | 0.0062 | 0.9907 | 0.0140 | 0.2809 | 0.0133 | 0.3394 | 0.0239 | 0.0066 | 0.0418 | 0.0000 | 0.0660 | 0.0000 |
| avg_word_length | 0.0067 | 0.9757 | 0.0081 | 0.8902 | 0.0091 | 0.8021 | 0.0101 | 0.6875 | 0.0178 | 0.0841 | 0.0269 | 0.0014 |
| sentence_count | 0.0110 | 0.5806 | 0.0088 | 0.8335 | 0.0121 | 0.4569 | 0.0231 | 0.0096 | 0.0327 | 0.0000 | 0.0495 | 0.0000 |
| unique_words_ratio | 0.0111 | 0.5689 | 0.0179 | 0.0812 | 0.0113 | 0.5457 | 0.0147 | 0.2301 | 0.0524 | 0.0000 | 0.0851 | 0.0000 |
| punctuation_ratio | 0.0092 | 0.7912 | 0.0203 | 0.0325 | 0.0182 | 0.0729 | 0.0184 | 0.0677 | 0.0512 | 0.0000 | 0.1063 | 0.0000 |
| avg_sentence_length | 0.0139 | 0.2888 | 0.0178 | 0.0841 | 0.0113 | 0.5457 | 0.0172 | 0.1038 | 0.0238 | 0.0069 | 0.0389 | 0.0000 |
| vocab_richness | 0.0052 | 0.1300 | 0.0077 | 0.0055 | 0.0100 | 0.0001 | 0.0177 | 0.0000 | 0.0492 | 0.0000 | 0.7523 | 0.0000 |
| short_word_ratio | 0.0079 | 0.9139 | 0.0120 | 0.4676 | 0.0098 | 0.6994 | 0.0100 | 0.6994 | 0.0161 | 0.1497 | 0.0140 | 0.2809 |
| long_word_ratio | 0.0105 | 0.6399 | 0.0197 | 0.0413 | 0.0087 | 0.8436 | 0.0072 | 0.9578 | 0.0130 | 0.3667 | 0.0303 | 0.0002 |
| Average | 0.0091 | 0.6533 | 0.0140 | 0.3019 | 0.0115 | 0.4810 | 0.0158 | 0.3069 | 0.0331 | 0.0675 | 0.1299 | 0.0314 |
| DateDetectionMIA | 0.482 | | 0.500 | | 0.504 | | 0.485 | | 0.509 | | 0.472 | |
| GreedyRareWordMIA | 0.503 | | 0.501 | | 0.498 | | 0.508 | | 0.503 | | 0.517 | |
| BagOfWordsMIA | 0.490 | | 0.500 | | 0.514 | | 0.517 | | 0.536 | | 0.542 | |

Table 1: Distribution shifts on **Pile** across deduplication strengths (100% to 10%). For each strength, we report: (1) KS tests with test statistics and p-values, and (2) the AUROC scores of blind baselines.

| Domains | Reddit | | Wiki | | C4* | | Falcon* | |
|---|---|---|---|---|---|---|---|---|
| Features | KS | p-value | KS | p-value | KS | p-value | KS | p-value |
| word_count | 0.0093 | 0.7802 | 0.0206 | 0.5599 | 0.0933 | 0.0000 | 0.0761 | 0.0000 |
| avg_word_length | 0.0255 | 0.0030 | 0.0227 | 0.4339 | 0.0256 | 0.0409 | 0.0428 | 0.0002 |
| sentence_count | 0.0139 | 0.2888 | 0.0198 | 0.6093 | 0.0832 | 0.0000 | 0.0813 | 0.0000 |
| unique_words_ratio | 0.0162 | 0.1449 | 0.0150 | 0.8933 | 0.0731 | 0.0000 | 0.0595 | 0.0000 |
| punctuation_ratio | 0.0314 | 0.0001 | 0.0141 | 0.9303 | 0.0353 | 0.0012 | 0.0516 | 0.0000 |
| avg_sentence_length | 0.0147 | 0.2301 | 0.0306 | 0.1272 | 0.0635 | 0.0000 | 0.0322 | 0.0119 |
| short_word_ratio | 0.0149 | 0.2169 | 0.0237 | 0.3790 | 0.0309 | 0.0069 | 0.0476 | 0.0000 |
| long_word_ratio | 0.0208 | 0.0264 | 0.0155 | 0.8664 | 0.0183 | 0.2698 | 0.0393 | 0.0010 |
| text_length | 0.0088 | 0.8335 | 0.0204 | 0.5715 | 0.0924 | 0.0000 | 0.0752 | 0.0000 |
| DateDetectionMIA | 0.500 | | 0.494 | | 0.514 | | 0.524 | |
| GreedyRareWordMIA | 0.498 | | 0.497 | | 0.501 | | 0.502 | |
| BagOfWordsMIA | 0.524 | | 0.501 | | 0.587 | | 0.575 | |

Table 2: Distribution shift analysis across **Dolma** domains using 70% intra-deduplication threshold. Domains marked with * are excluded from the final testbed due to significant distribution shifts.

**Pythia/Pile.** Our analysis covers both the globally sampled datasets across multiple deduplication strengths ($d \in 100, 90, 70, 50, 30, 10$) and the domain-level datasets.

*Global-level datasets.* We observe larger distribution shifts as deduplication strength increases (lower $d$), indicated by both the increasing performance of blind baselines and higher KS statistics with lower $p$-values (Table 1). This highlights the inherent tension between deduplication and low distribution shifts. **Given this trade-off, we recommend researchers report MIA success across multiple deduplication levels.** Following Anil et al. (2023), we establish $d = 70$ as the default deduplication strength, where we observe negligible distribution shift between members and non-members. We further recommend calibrating MIA performance against the performance of the strongest blind baseline, as opposed to random accuracy/AUROC of 50%).

Our results also indicate that KS tests are generally more sensitive at the population level compared to blind baselines. Notably, a $p$-value below 0.05 does not automatically disqualify a dataset, particularly when it had a small KS test statistic and that $p$-values also accounts for the sample size. We thus set a KS statistic threshold of 0.04 for the inclusion of a dataset.

*Domain-level datasets.* Under the default dedup strength of 70%, we found that of the 17 available domains from the Pile, 3 exhibited severe member/non-member distribution shifts (NIH Ex-Porter, DM Mathematics, and USPTO Backgrounds), and another 5 have less 3,000 samples post-dudplication. This leaves 9 workable domains from which we release verified domain-level datasets. See See Appendix F for full results, including distribution shifts on domain-specific datasets.

**Olmo/Dolma.** Olmo/Dolma presents unique challenges as Dolma (training tokens) and Paloma (evaluation tokens) differ in both collection methodology and domain composition. Our global random sampling revealed substantial distribution shifts, as shown in Table 2. Paloma also has generally smaller per-domain sample sizes, which limited the availability of distribution-matched data. After analyzing four common domains with over 1,500 samples in Paloma, we identified Reddit and Wiki as having acceptable levels of distribution shift.

**GPT-2/FineWeb.** As a fully controlled setting, we trained a GPT-2-Large model from scratch on FineWeb-Edu Penedo et al. (2024) with explicit train-validation splits. Our analysis shows distribution shift patterns similar to the Pythia/Pile setting. See Appendix F for details.

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

## A    CONCLUDING REMARKS

We introduced a rigorous testbed for evaluating membership inference attacks on LLM pre-training sequences, addressing key gaps in prior work. Our framework ensures uniform and efficient sampling from pre-training shards, large-scale deduplication, and statistical validation to detect distribution shifts across different deduplication strengths. By standardizing a framework for building MIA evaluation datasets, we hope to facilitate future research in this critical area.

## B    RELATED WORK

Our work is not the first to provide evaluation data for MIAs against pre-trained LLMs. Duan et al. (2024) reported temporal shifts in the WikiMIA dataset (Shi et al., 2023), and in turn proposed the MIMIR benchmark which included Temporal arXiv and Temporal Wiki datasets; Temporal arXiv/Wiki are subsequently reported by Das et al. (2024) to still be susceptible to model-blind attacks like bag-of-words classification. MIMIR also included a Pile random sample, though the uniform randomness was implemented at the level of pre-training shards rather than documents, and distribution shifts were only examined with existing (ineffective) MIA methods without rigorous statistical tests across different deduplication levels. Meeus et al. (2024a) proposed using books before/after training cut-off as members/non-members, and was similarly shown by Das et al. (2024) to have severe distribution shifts. As follow-up, Meeus et al. (2024d) released a Pile-based arXiv paper split; the latter has low sample size and is similarly not statistically tested across deduplication strengths as done in our work. Meeus et al. (2024c) considered injecting synthetic strings to training data and use similar, held-out strings as non-members, but the unrealistic nature of these strings may imply that MIA evaluation may not reflect real-world performance. Our work aims to fill these gaps.

## C    IMPLEMENTATION DETAILS

**Other model/dataset pairs.**   We explored other model/dataset candidate pairs, including the Molmo/Pixmo (vison-language model), but found them unsuitable for our purposes, either due to limited number of available examples or observing large member/non-member distribution shifts after deduplication.

**Indices for random sampling.**   During preprocessing, we scan each shard once to index the position and count of newline characters, tracking line numbers across different domains and maintaining domain-specific data counts per shard. This stage leverages multi-threading to accelerate processing, scaling efficiently with available memory and CPU cores. When randomly sampling from the sharded dataset, we use the total line counts per shard to determine the sampling distribution across shards, then employ randint to select specific lines within each shard. Thanks to our indexing approach, we can access any line in $O(1)$ time complexity. Using this method, we successfully preprocessed the entire Dolma dataset—11TB of data or 3 trillion tokens—in just 2 days using 16 threads on a machine with 300GB of memory. Once preprocessing is complete, our index enables repeated $O(1)$ time sampling operations. To facilitate broader research use, we will release our preprocessed cache, allowing users to perform immediate random sampling without repeating the preprocessing step.

**Document-level testbed.**   Our evaluation sets are *document-level* by design, meaning that every member and non-member example in the dataset corresponds to a complete document in the original, untokenized dataset (e.g., one row of Pile). This design better reflects real-world usage (as opposed to having fixed-length member and non-member strings).

**Global- and domain-specific datasets.**   Beyond sampling members/non-members globally, we also assess MIA success across domain-specific subsets. This involves examining the metadata of each document in the pre-training dataset and group them into domains such as Reddit, Stack Exchange, and Wikipedia. This approach gives more fine-grained insights into MIA performance. In total, our pipeline (fig. 1) yields 9 validated domains from Pile and 2 from Dolma. See Appendix D for details.

## D  DOMAIN-SPECIFIC MEMBERS AND NON-MEMBERS

**Available domains.**  We list all the available domains from which we draw evaluation members and non-members:

- **Pile**: ArXiv, Pile-CC, StackExchange, Wikipedia, FreeLaw, Github, HackerNews, PubMed Abstracts, PubMed Central.
- **Dolma**: Reddit and Wiki.

## E  TEXTUAL FEATURES FOR TWO-SAMPLE KOLMOGOROV–SMIRNOV (KS) TESTS

The nine different textual features that we extracted for our KS tests are:

1. **Word Count**: The total number of words in the text, calculated by counting all sequences of words.

2. **Average Word Length**: The mean length of all words in the text, providing a measure of vocabulary complexity.

3. **Sentence Count**: The total number of sentences in the text, determined by counting sentence-ending punctuation marks (.!?).

4. **Unique Words Ratio**: The proportion of unique words to total words, calculated as the number of distinct words divided by the total word count.

5. **Punctuation Ratio**: The density of punctuation marks in the text, computed as the count of punctuation characters divided by the total character count.

6. **Average Sentence Length**: The mean number of words per sentence, indicating typical sentence complexity.

7. **Short Word Ratio**: The proportion of words that are 4 characters or shorter, representing the usage of simple or common words.

8. **Long Word Ratio**: The proportion of words that are 7 characters or longer, indicating the usage of complex or specialized vocabulary.

9. **Text Length**: The total number of characters in the text, including spaces and punctuation marks.

## F  ADDITIONAL RESULTS

**Dataset statistics.**  We present our testbed statistics in Table 3. For each testbed, we generally aim to balance the number of member and non-member samples. While our target sampling size is shown in the table, some domain-specific cases contain fewer samples (e.g. approximately 4,000) due to data availability constraints after deduplication. In these cases, we maintain the target size for the member set while including all available deduplicated samples in the non-member set.

| Dataset Characteristics | Pythia/Pile | Olmo/Dolma | GPT-2/FineWeb |
|---|---|---|---|
| Training Dataset Size (docs) | 177M | 3.395B | 16M |
| Global Testbed Size (docs) | 50,000 | - | 50,000 |
| Available Domain-specific Testbeds | 9 | 2 | - |
| Per-Domain Testbed Size (docs) | 20,000 | 20,000 | - |
| Model Size (parameters) | 1.4B | 7B | 762M |

Table 3: Summary statistics of our constructed testbeds across three experimental settings. For each setting, we report the size of the original training corpus, the number of documents in our global and domain-specific testbeds, the number of domains where distribution-matched data was available, and the size of the corresponding language model. A dash (-) indicates that the particular characteristic is not applicable for that setting.

| Domains | Pile-CC | | StackExchange | | Wikipedia | | ArXiv | | DM Mathematics* | |
|---|---|---|---|---|---|---|---|---|---|---|
| Features | KS | p-value | KS | p-value | KS | p-value | KS | p-value | KS | p-value |
| word_count | 0.0125 | 0.0969 | 0.0080 | 0.5419 | 0.0139 | 0.0430 | 0.0207 | 0.0754 | 0.0770 | 0.0000 |
| avg_word_length | 0.0101 | 0.2756 | 0.0059 | 0.8732 | 0.0084 | 0.4941 | 0.0201 | 0.0919 | 0.0726 | 0.0000 |
| sentence_count | 0.0119 | 0.1289 | 0.0047 | 0.9795 | 0.0089 | 0.4170 | 0.0131 | 0.5287 | 0.0301 | 0.0100 |
| unique_words_ratio | 0.0106 | 0.2251 | 0.0103 | 0.2414 | 0.0119 | 0.1194 | 0.0161 | 0.2743 | 0.0803 | 0.0000 |
| punctuation_ratio | 0.0043 | 0.9941 | 0.0121 | 0.1057 | 0.0085 | 0.4667 | 0.0258 | 0.0124 | 0.0782 | 0.0000 |
| avg_sentence_length | 0.0080 | 0.5611 | 0.0051 | 0.9582 | 0.0139 | 0.0432 | 0.0161 | 0.2790 | 0.0204 | 0.1739 |
| short_word_ratio | 0.0106 | 0.2242 | 0.0086 | 0.4482 | 0.0103 | 0.2406 | 0.0182 | 0.1606 | 0.0743 | 0.0000 |
| long_word_ratio | 0.0095 | 0.3406 | 0.0077 | 0.5960 | 0.0083 | 0.5037 | 0.0157 | 0.3046 | 0.0668 | 0.0000 |
| text_length | 0.0127 | 0.0866 | 0.0077 | 0.6004 | 0.0137 | 0.0490 | 0.0224 | 0.0437 | 0.0001 | 1.0000 |
| Average | 0.0100 | 0.3259 | 0.0078 | 0.5938 | 0.0109 | 0.2641 | 0.0187 | 0.1967 | 0.0555 | 0.1315 |
| DateDetectionMIA | 0.502 | | 0.501 | | 0.493 | | 0.501 | | 0.514 | |
| BagOfWordsMIA | 0.504 | | 0.491 | | 0.506 | | 0.505 | | 0.533 | |
| GreedyRareWordMIA | 0.501 | | 0.500 | | 0.501 | | 0.500 | | 0.509 | |

Table 4: Domain-specific distribution shifts for **Pile** domains at 70% deduplication threshold. Domains marked with * are excluded from the final testbed due to significant distribution shifts.

**Additional domains for Pythia/Pile.** For the Pythia/Pile setting, we analyzed additional domains beyond the four non-shift and one shifted domain discussed in Section 3. We deferred results on domain-specific datasets from the main text, and they can be found in Table 4 and Table 5. Not all domain-specific datasets in Pile are usable after deduplication; for example, as shown in Table 5, after applying a 70% deduplication threshold, we found severe distribution shifts in the NIH Exporter and USPTO Backgrounds domains, rendering them unsuitable for inclusion in our benchmark.

| Feature | NIH ExPorter* | | FreeLaw | | Github | | HackerNews | | PubMed Abstracts | | PubMed Central | | USPTO Backgrounds* | |
|---|---|---|---|---|---|---|---|---|---|---|---|---|---|---|
| | KS | p-val | KS | p-val | KS | p-val | KS | p-val | KS | p-val | KS | p-val | KS | p-val |
| word_count | 0.0615 | 0.0000 | 0.0182 | 0.0278 | 0.0062 | 0.8718 | 0.0200 | 0.2128 | 0.0072 | 0.6736 | 0.0192 | 0.0098 | 0.0235 | 0.0004 |
| avg_word_length | 0.0160 | 0.6358 | 0.0103 | 0.5041 | 0.0197 | 0.0018 | 0.0250 | 0.0598 | 0.0076 | 0.6216 | 0.0160 | 0.0508 | 0.0528 | 0.0000 |
| sentence_count | 0.0522 | 0.0000 | 0.0169 | 0.0494 | 0.0089 | 0.4640 | 0.0186 | 0.2870 | 0.0069 | 0.7341 | 0.0188 | 0.0121 | 0.0294 | 0.0000 |
| unique_words_ratio | 0.0479 | 0.0001 | 0.0126 | 0.2570 | 0.0120 | 0.1461 | 0.0162 | 0.4552 | 0.0062 | 0.8412 | 0.0152 | 0.0704 | 0.0235 | 0.0004 |
| punctuation_ratio | 0.0338 | 0.0145 | 0.0070 | 0.9044 | 0.0298 | 0.0000 | 0.0181 | 0.3146 | 0.0070 | 0.7132 | 0.0193 | 0.0091 | 0.0386 | 0.0000 |
| avg_sentence_length | 0.0315 | 0.0277 | 0.0111 | 0.3991 | 0.0061 | 0.8871 | 0.0214 | 0.1543 | 0.0082 | 0.5151 | 0.0135 | 0.1446 | 0.0384 | 0.0000 |
| short_word_ratio | 0.0195 | 0.3853 | 0.0078 | 0.8228 | 0.0150 | 0.0342 | 0.0205 | 0.1883 | 0.0085 | 0.4624 | 0.0175 | 0.0247 | 0.0407 | 0.0000 |
| long_word_ratio | 0.0193 | 0.3931 | 0.0054 | 0.9920 | 0.0184 | 0.0043 | 0.0267 | 0.0371 | 0.0051 | 0.9611 | 0.0149 | 0.0799 | 0.0495 | 0.0000 |
| text_length | 0.0634 | 0.0000 | 0.0194 | 0.0155 | 0.0058 | 0.9155 | 0.0197 | 0.2255 | 0.0074 | 0.6480 | 0.0181 | 0.0173 | 0.0249 | 0.0001 |
| DateDetectionMIA (AUROC) | 0.503 | | 0.488 | | 0.485 | | 0.492 | | 0.497 | | 0.494 | | 0.495 | |
| BagOfWordsMIA (AUROC) | 0.512 | | 0.507 | | 0.500 | | 0.506 | | 0.491 | | 0.506 | | 0.532 | |
| GreedyRareWordMIA (AUROC) | 0.502 | | 0.500 | | 0.502 | | 0.499 | | 0.501 | | 0.501 | | 0.501 | |

Table 5: Additional domain-specific distribution shift analysis for the remaining 7 domains at 70% deduplication threshold. Domains marked with * are excluded from the final testbed due to significant distribution shifts.

**GPT-2/FineWeb.** We trained a GPT-2-Large model *from scatch* using LLM.c (Karpathy, 2024) on an explicit train-validation split of FineWeb-Edu Penedo et al. (2024). Since FineWeb provides only coarse-grained document metadata, we focus on the global-level datasets at different deduplication strengths, similar to the Pythia/Pile setting. Distribution shifts are evaluated in Table 6 in Appendix F, showing patterns similar to the Pythia/Pile setting. We release model checkpoints after 1/2/3 epochs to facilitate MIA evaluations accounting for data repetition. indicating Common Crawl version source Beyond these three settings, we also attempted to construct a testbed using Molmo/Pixmo. However, we observed significant distribution shifts immediately after random sampling, suggesting non-negligible inherent distribution differences between the training and validation sets.

# G ANALYSIS

**Dataset-level vs. instance-level evaluation.** The relationship between dataset-level distribution tests and instance-level membership inference requires careful consideration. While passing dataset-level tests (such as the KS test) is necessary, it does not invalidate the dataset's utility for evaluating membership inference attacks. Rather, successful dataset-level tests ensure that any observed membership inference performance stems from genuine membership signals rather than artifacts of distribution shift. This separation enables us to disentangle distribution inference effects from true membership inference capabilities, providing a more reliable evaluation framework for MIA research.

| Duplicate Strength | 100% | | 90% | | 70% | | 50% | | 30% | | 10% | |
|---|---|---|---|---|---|---|---|---|---|---|---|---|
| Features | KS | p-value | KS | p-value | KS | p-value | KS | p-value | KS | p-value | KS | p-value |
| word_count | 0.0053 | 0.4724 | 0.0080 | 0.0893 | 0.0101 | 0.0205 | 0.0078 | 0.1418 | 0.0102 | 0.0268 | 0.0119 | 0.0119 |
| avg_word_length | 0.0038 | 0.8701 | 0.0042 | 0.7795 | 0.0066 | 0.2830 | 0.0062 | 0.3852 | 0.0091 | 0.0646 | 0.0200 | 0.0000 |
| sentence_count | 0.0054 | 0.4485 | 0.0066 | 0.2472 | 0.0074 | 0.1705 | 0.0063 | 0.3577 | 0.0093 | 0.0571 | 0.0111 | 0.0231 |
| unique_words_ratio | 0.0056 | 0.4027 | 0.0060 | 0.3450 | 0.0048 | 0.6712 | 0.0082 | 0.1076 | 0.0145 | 0.0003 | 0.0203 | 0.0000 |
| punctuation_ratio | 0.0113 | 0.0033 | 0.0071 | 0.1747 | 0.0126 | 0.0016 | 0.0211 | 0.0000 | 0.0301 | 0.0000 | 0.0518 | 0.0000 |
| avg_sentence_length | 0.0072 | 0.1447 | 0.0086 | 0.0534 | 0.0107 | 0.0123 | 0.0081 | 0.1207 | 0.0067 | 0.3019 | 0.0058 | 0.5805 |
| short_word_ratio | 0.0041 | 0.7931 | 0.0049 | 0.5961 | 0.0074 | 0.1712 | 0.0065 | 0.3205 | 0.0099 | 0.0353 | 0.0212 | 0.0000 |
| long_word_ratio | 0.0029 | 0.9853 | 0.0027 | 0.9937 | 0.0060 | 0.3991 | 0.0049 | 0.6878 | 0.0070 | 0.2603 | 0.0158 | 0.0002 |
| text_length | 0.0057 | 0.3852 | 0.0083 | 0.0706 | 0.0098 | 0.0282 | 0.0077 | 0.1560 | 0.0105 | 0.0204 | 0.0106 | 0.0348 |
| Average | 0.0057 | 0.501 | 0.0063 | 0.372 | 0.0084 | 0.195 | 0.0085 | 0.253 | 0.0119 | 0.085 | 0.0187 | 0.072 |
| DateDetectionMIA | 0.501 | | 0.500 | | 0.507 | | 0.501 | | 0.508 | | 0.504 | |
| GreedyRareWordMIA | 0.499 | | 0.500 | | 0.500 | | 0.500 | | 0.499 | | 0.500 | |
| BagOfWordsMIA | 0.500 | | 0.509 | | 0.514 | | 0.513 | | 0.511 | | 0.537 | |

Table 6: Distribution shift analysis on FineWeb across deduplication thresholds. Average KS statistics, p-values, and blind baseline performance are provided for each threshold.

**Discussions on deduplication strength.** The inherent ambiguity of MIA, as discussed in Duan et al. (2024), complicates duplicate definition through two key challenges: (1) high $n$-gram overlap between non-members and the training corpus creates "partial duplicates" that contain training data signals, blurring the member/non-member distinction, and (2) stronger deduplication can introduce more severe distribution shifts, potentially conflating membership inference with distribution inference (Suri & Evans, 2022). This presents a fundamental trade-off between maintaining distributional consistency and ensuring clear membership distinction. As a testbed, we aim to balance these competing objectives. To better understand how deduplication strength influences this trade-off, we conduct comprehensive distribution shift analyses across a spectrum of strength values.

# H DISTRIBUTION SHIFT DETECTION METHODOLOGY

**Blind baselines.** Recent work by Das et al. (2024); Meeus et al. (2024b) introduced "blind" baselines that attempt to distinguish members vs. non-members *without* accessing the model. Because of widespread distribution shifts in existing evaluation data, these baselines outperform existing MIA methods by a large margin. On an ideal evaluation dataset, they should perform (near-)randomly; we re-implement these baselines as an initial check of distribution shifts. In our analysis, we consider three such blind baselines: date detection, greedy rare word detection, and bag-of-words classification. (We refer the reader to Das et al. (2024); Meeus et al. (2024b) for more details.)

**Two-sample hypothesis tests.** While blind baselines detect coarse distribution shifts, we need additional rigor via statistical methods. We extract a comprehensive set of textual features from the documents, including: document word count, average word length, sentence count, unique words ratio, punctuation ratio, average sentence length, vocab richness, short word ratio, and long word ratio (see Appendix E for details). We then apply the Kolmogorov-Smirnov (KS) test (Massey, 1951) to these features from the members and non-members as a two-sample hypothesis test. The KS test statistic measures the magnitude of distribution shifts, and the $p$-value indicates confidence for the given amount of samples. The KS tests are much more powerful (and sensitive) than the blind baselines (§3); we use them as an important indicator of data quality, though strict passing of these tests is not always necessary for our testbed to be useful for MIA evaluation.

