# OpenReview forum: "A Missing Testbed for LLM Pre-Training Membership Inference Attacks"
_ICLR.cc/2025/Workshop/BuildingTrust — BuildingTrust_

### Official Review · Reviewer_x5LU · 2025-02-24
**Review of "A Missing Testbed for LLM Pre-Training Membership Inference Attacks"**

**Rating:** 7
**Confidence:** 3

**Review:**

### Strength
This paper presents a standardized testbed for evaluating membership inference attacks (MIAs) on LLM pre-training data, addressing key issues in data bias, distribution shifts, and unreliable evaluation methods. It introduces true uniform sampling, large-scale deduplication, and Kolmogorov-Smirnov statistical tests to improve dataset quality and prevent misleading MIA results. The work is highly reproducible, releasing open-source datasets, models, and evaluation tools, making it a valuable contribution to MIA research.

### Weakness
However, the paper lacks real-world attack case studies, focusing mainly on dataset validation without directly evaluating state-of-the-art MIAs on the proposed testbed. Additionally, computational cost and scalability are not fully addressed, raising concerns about its feasibility for large-scale models. The study also relies heavily on statistical tests without deep analysis of MIA performance variations under different dataset conditions.

---

### Official Review · Reviewer_wQaP · 2025-02-27
**Important direction, but structure of the paper should be improved (or submitted to a long-track)**

**Rating:** 3
**Confidence:** 2

**Review:**

The authors propose a testbed for membership inference attacks that addresses some gaps in the current evaluations.

Strengths:
* The introduced testbed is claimed to possess three important properties that address previous gaps in the literature, such as true uniform sampling of member/non-member sequences.
* This paper gives important insights on how to correctly evaluate MIAs.

Weaknesses:
* Structure: the paper reads like a full paper, where important sections were moved to the appendix due to 4 pages limitation (including Conclusion, Related work, some results and analysis, etc). This really should have been submitted to the Long-papers track.
* I am not familiar with MIAs, and this is hard for me to judge the technical solidness of the paper. Furthermore, as a reader outside of the field, it was really hard for me to follow the paper, since lots of important pieces were hidden in the appendix (like missing the context of the work, which is described in the appendix). Due to the issues with the paper structure, I’d recommend rejection, however, I am not certain in my evaluation as non-expert.

---

### Official Review · Reviewer_bbRN · 2025-02-28
**Testbed and dataset for Membership Inference Attacks**

**Rating:** 7
**Confidence:** 3

**Review:**

# Review

The authors introduce a testbed and dataset for Membership Inference Attacks (MIA). They provide both global- and domain-level datasets through uniform sampling based on fully open model/dataset pairs and open-source a modular codebase.

## Strengths

1. High-quality data cleaning methods: The authors extract textual features and conduct Kolmogorov-Smirnov (KS) tests. Additionally, they implement blind baselines to detect significant distribution shifts between members and non-members.
2. Open-source datasets: The authors provide global- and domain-level datasets derived from fully open model/dataset pairs.
3. Extensibility: The same approach can be applied to sensitive industry data processing, which is beneficial for data security.

## Weaknesses
The effects of different dataset constructions are not discussed. Additionally, the sensitivity of MIA may vary across different model architectures.

---

### Decision · Program_Chairs · 2025-03-04

Accept